# Modelling the synergistic effect of bacteriophage and antibiotics on bacteria: Killers and drivers of resistance evolution

**Quentin J. Leclerc**[1,2,3¤]*, **Jodi A. Lindsay**[3], **Gwenan M. Knight**[1,2]

**1** Centre for Mathematical Modelling of Infectious Diseases, Department of Infectious Disease Epidemiology, Faculty of Epidemiology & Population Health, London School of Hygiene & Tropical Medicine, London, United Kingdom, **2** Antimicrobial Resistance Centre, London School of Hygiene and Tropical Medicine, London, United Kingdom, **3** Institute for Infection & Immunity, St George's University of London, London, United Kingdom

¤ Current address: Epidemiology and Modelling of Bacterial Escape to Antimicrobials, Institut Pasteur, Paris, France

\* p1807380@sgul.ac.uk, quentin.leclerc@pasteur.fr

**Data Availability Statement:** All relevant data and code are hosted in a GitHub repository: https://github.com/qleclerc/phage_antibiotic_dynamics.

## Abstract

Bacteriophage (phage) are bacterial predators that can also spread antimicrobial resistance (AMR) genes between bacteria by generalised transduction. Phage are often present alongside antibiotics in the environment, yet evidence of their joint killing effect on bacteria is conflicted, and the dynamics of transduction in such systems are unknown. Here, we combine *in vitro* data and mathematical modelling to identify conditions where phage and antibiotics act in synergy to remove bacteria or drive AMR evolution. We adapt a published model of phage-bacteria dynamics, including transduction, to add the pharmacodynamics of erythromycin and tetracycline, parameterised from new *in vitro* data. We simulate a system where two strains of *Staphylococcus aureus* are present at stationary phase, each carrying either an erythromycin or tetracycline resistance gene, and where multidrug-resistant bacteria can be generated by transduction only. We determine rates of bacterial clearance and multidrug-resistant bacteria appearance, when either or both antibiotics and phage are present at varying timings and concentrations. Although phage and antibiotics act in synergy to kill bacteria, by reducing bacterial growth antibiotics reduce phage production. A low concentration of phage introduced shortly after antibiotics fails to replicate and exert a strong killing pressure on bacteria, instead generating multidrug-resistant bacteria by transduction which are then selected for by the antibiotics. Multidrug-resistant bacteria numbers were highest when antibiotics and phage were introduced simultaneously. The interaction between phage and antibiotics leads to a trade-off between a slower clearing rate of bacteria (if antibiotics are added before phage), and a higher risk of multidrug-resistance evolution (if phage are added before antibiotics), exacerbated by low concentrations of phage or antibiotics. Our results form hypotheses to guide future experimental and clinical work on the impact of phage on AMR evolution, notably for studies of phage therapy which should investigate varying timings and concentrations of phage and antibiotics.

**Funding:** Q.J.L was supported by a studentship from the Medical Research Council Intercollegiate Doctoral Training Program (MR/N013638/1, https://mrc.ukri.org/). G.M.K was supported by a Skills Development Fellowship from the Medical Research Council (MR/P014658/1, https://mrc.ukri.org/). G.M.K and J.A.L were supported by a Research Grant from the Medical Research Council (MR/P028322/1, https://mrc.ukri.org/). The funders played no role in the study design, data collection and analysis, decision to publish, or preparation of the manuscript.

**Competing interests:** The authors have declared that no competing interests exist.

## Author summary

Bacteriophage ("phage") are viruses that can infect and kill bacteria, but also natural drivers of antimicrobial resistance (AMR) evolution by transduction, when they carry non-phage DNA between bacteria, including AMR genes. Phage are often present alongside antibiotics in the environment and in humans, yet the joint dynamics of phage, antibiotics, and bacteria are unclear. Using laboratory work and mathematical modelling, we show for the first time that depending on timing and concentration, phage and antibiotics can either work together to kill bacteria faster, or phage can generate multidrug-resistant bacteria by transduction which are then selected for by antibiotics. This may be particularly important in the context of phage therapy, where phage are used to treat bacterial infections, often alongside antibiotics. Our conclusions highlight the urgent need for clinical and laboratory work to quantify the currently unknown contribution of phage to AMR evolution in humans. Otherwise, we may be missing opportunities to reduce the global public health burden of AMR.

## Introduction

Bacteriophage (phage) are major bacterial predators and the most common organisms on the planet [1]. Phage are often present alongside antibiotics, naturally or in the context of antibacterial treatment [2–4], yet reports on their combined effect on bacteria are conflicting. Some previous studies showed that phage and antibiotics work synergistically to clear bacteria [5–8], while others have demonstrated that antibiotics reduce phage production [9,10]. This interaction has been extensively studied in biofilms to show that adding phage to disrupt the biofilm before adding antibiotics leads to the greatest synergistic killing [11–13]. However, whilst there is some work suggesting a similar timing-dependent synergistic killing effect on planktonic bacteria [6–8], this evidence is limited as only a few timings and concentrations of phage and antibiotics have been explored experimentally. This is further complicated by the fact that phage are also major drivers of bacterial evolution via horizontal gene transfer by transduction [14,15], which can notably contribute to antimicrobial resistance (AMR) spread [16,17]. If multidrug-resistant bacteria are generated by transduction, antibiotics present in the same environment may act as a synergistic selective pressure to increase their prevalence, yet to our knowledge the dynamics of transduction have not yet been investigated in such systems [18].

There are two types of transduction: specialised and generalised [19,20]. Specialised transduction occurs when a prophage sometimes picks up adjacent bacterial DNA upon excision from the bacterial chromosome, at the end of the lysogenic cycle. We also note lateral transduction as an important variation of specialised transduction, where phage packaging occurs before the prophage is excised from the bacterial chromosome, leading to transfer of DNA located further downstream of the prophage [21]. On the other hand, generalised transduction occurs during phage replication, when random non-phage DNA is packaged instead of phage DNA in a new phage particle. The resulting transducing phage is released upon bacterial lysis, and injects this DNA in another bacterium. Generalised transduction is likely the most important type of transduction in the context of AMR, as it can lead to the horizontal transfer of any genetic material contained in a bacterium (including plasmids, major vectors of AMR genes) [14].

A major bacterial pathogen often exposed to phage is *Staphylococcus aureus*, which at any given time is colonising approximately 20% of humans [22]. Previous work suggests that all *S.*

*aureus* carry integrated prophage [23], and at least 50% of individuals colonised by *S. aureus* also carry free phage capable of generalised transduction [24], the main mechanism of horizontal gene transfer for *S. aureus* [23]. This is particularly relevant for methicillin-resistant *S. aureus* (MRSA), a group of *S. aureus* present in both the ESKAPE list and the World Health Organization priority list of antibiotic-resistant bacteria due to its large clinical burden [25–27]. *In vitro*, the generation rate of generalised transducing MRSA phage carrying an AMR gene has been estimated to be approximately one per $10^8$ new phage produced, sufficient to consistently generate bacteria resistant to multiple antibiotics in less than 24h [28], and *in vivo* transduction rates are likely to be even higher [29].

Understanding the dynamics of bacteria, antibiotics, phage and transduction is especially important in the context of phage therapy, which aims to use phage as antibacterial agents, generally in combination with antibiotics [4]. Phage therapy is currently investigated as a solution to counter the threat of AMR, with ongoing clinical trials against MRSA infection [6,30–34]. Phage therapy guidelines recommend that only phage with a limited ability to perform transduction should be used [19,35,36], yet to our knowledge there is currently no technique to prevent phage from accomplishing generalised transduction, which is fundamentally a mispackaging and thus a biological error similar to a mutation. Hence, as previous reviews have highlighted, it is essential to explore the importance of this mechanism, and identify conditions under which it could affect the outcome of therapy and lead to multidrug-resistance evolution [18,35–37].

In this study, we aim to investigate the potential combined effect of antibiotics and phage capable of generalised transduction on bacteria. Making conclusions about this multidimensional chequerboard space of potential combinations and timings is difficult when using data from time-consuming, single-scenario, *in vitro* experiments. Instead, we generate hypotheses to guide future experimental work by simulating these conditions. We extend a previously published mathematical model of only phage-bacteria dynamics, including generalised transduction, to newly incorporate the effect of antibiotics on bacteria [28]. This model is parameterised using *in vitro* data from the same environment and set of conditions as when it was originally developed [28], making it a reliable tool to infer the dynamics governing this system.

We hypothesise that, depending on the timing and concentration at which they are added, phage and antibiotics can either act synergistically to eradicate bacteria, or to create and select for multidrug-resistant bacteria. We explore this in our model and generate guidelines to minimise the risk of generating double-antibiotic-resistant *S. aureus* when two single-resistant strains are exposed to antibiotics and phage, whilst maximising bacterial eradication.

## Materials and methods

### Laboratory methods

**Bacterial strains and phage.**   Two *Staphylococcus aureus* strains were obtained from the Nebraska transposon library in the MRSA USA300 background [38]. These were NE327, with the *ermB* gene conferring resistance to erythromycin, and NE201KT7, a modified NE201 strain with a kanamycin resistance cassette replacing the *ermB* gene and a plasmid carrying the *tetK* gene conferring resistance to tetracycline. Horizontal gene transfer can only happen between these two strains via generalised transduction. When co-cultured with 80α phage, these give rise to double-resistant progeny (DRP) bacteria resistant to both erythromycin and tetracycline [28]. In the experiments conducted here, we used the double-resistant strain DRPET1, a DRP strain with a NE327 background generated during previous NE327, NE201KT7 and 80α co-cultures, containing both *ermB* and *tetK* genes [28]. Note that although 80α is a temperate phage, we have previously shown that lysogeny does not occur at a

detectable level in our experiments, with a lysogenic frequency detection limit of 3.3 x $10^{-8}$ per non-lysogenic bacteria after 24h of co-culture, and that specialised transduction is unlikely to be responsible for horizontal gene transfer, due to the location of *tetK* on a plasmid and a ~1Mbp distance between *ermB* and the 80α integration site on the bacterial chromosome [28]. Hence the only interactions we consider between phage and bacteria in our environment are lysis and generalised transduction.

### Time-kill curves

The growth conditions are identical to the ones previously used to generate data on lysis and transduction with these bacteria and phage [28]. Pre-cultures of each *S. aureus* bacterial strain (NE327, NE201KT7 and DRPET1) were separately generated overnight in 50mL conical tubes containing 10mL of brain-heart infusion broth (BHIB, Sigma, UK). Unless otherwise stated, liquid cultures were incubated in a warm shaking water bath (37˚C, 90 rpm). Each pre-culture was then diluted in phosphate-buffered saline (PBS) and mixed with 10mL of fresh BHIB in a 50mL conical tube to reach a starting concentration of $10^4$ colony-forming units (cfu)/mL. The new culture was incubated for 2h to allow the bacteria to reach log-growth phase, following standard protocol for time-kill experiments [39]. Erythromycin or tetracycline was then added to the culture, at a concentration of either 0 (control), 0.25, 0.5, 1, 2, 4, 8, 16, or 32 mg/L. At 0, 1, 2, 3, 4, 6 and 24h after antibiotic addition, 30μl were sampled from the incubated culture, diluted in PBS, and plated on plain brain-heart infusion agar. The plates were incubated overnight at 37˚C. Colonies on the plates were then counted to derive the concentration of bacteria in cfu/mL at the corresponding time point.

The experiment was repeated 3 times for each strain (NE327, NE201KT7 and DRPET1) and each antibiotic (erythromycin and tetracycline).

### Minimum inhibitory concentration

We measured the minimum inhibitory concentrations (MIC) of erythromycin and tetracycline for NE327, NE201KT7 and DRPET1 by microbroth dilution [40]. Briefly, pre-cultures of each strain were generated overnight in Mueller-Hinton broth (MHB-II, Sigma, UK). Antibiotic stocks were generated in 50% ethanol at concentrations doubling from 0.25 to 256 mg/L, and 10μL of each dilution were added to separate wells on a 96-well plate. The overnight pre-cultures were diluted to a concentration of $10^5$ cfu/mL, and 90μL were mixed with each antibiotic dilution in the 96-well plate. The plate was incubated overnight at 37˚C, and the MIC were then determined by eye, identifying the lowest concentration of antibiotic which did not allow bacterial growth (i.e. the contents of the well were not turbid).

### Modelling methods

All analyses were conducted in the R statistical analysis software [41]. The underlying code and data are available in a GitHub repository: https://github.com/qleclerc/phage_antibiotic_dynamics.

### Mathematical model description

We extended a previously published mathematical model of phage-bacteria dynamics, including generalised transduction [28]. This original model focused on identifying the best method to capture the interaction between bacteria and phage *in vitro*, in the absence of antibiotics. This model was previously parameterised using the same bacterial and phage strains as in this study. Our three *S. aureus* strains are represented: $B_E$ (erythromycin-resistant, corresponding

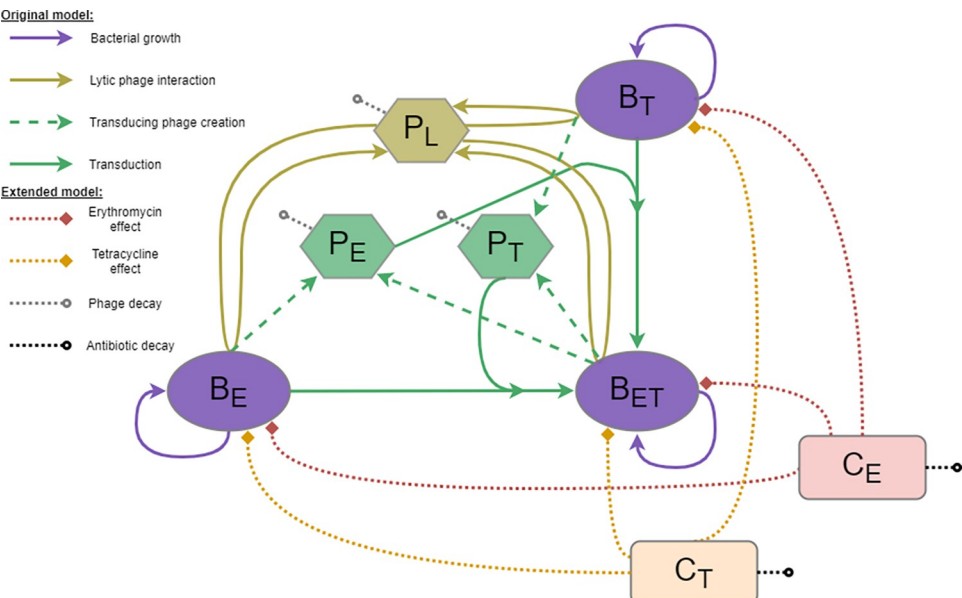

**Fig 1. Mathematical model diagram.** This model is an extension of our original model presented in [28], with the inclusion of antibiotic effects. Each bacteria strain ($B_E$ resistant to erythromycin, $B_T$ resistant to tetracycline, or $B_{ET}$ resistant to both) can replicate (purple). The lytic phage ($P_L$) multiply by infecting a bacterium and bursting it to release new phage (gold). This process can create transducing phage ($P_E$ or $P_T$) carrying a resistance gene (*ermB* or *tetK* respectively) taken from the infected bacterium (dashed, green). These transducing phage can then generate new double-resistant progeny ($B_{ET}$) by infecting the bacteria strain carrying the other resistance gene (solid, green). The antibiotics, erythromycin ($C_E$) and tetracycline ($C_T$), decrease the growth rate of each bacteria strain to varying extents, depending on their concentration and the resistance level of the strain (dotted, red and orange). Phage and antibiotics can decay at a fixed rate (dotted, grey and black).

to *in vitro* strain NE327), $B_T$ (tetracycline-resistant, corresponding to NE201KT7), and $B_{ET}$ (resistant to both erythromycin and tetracycline, corresponding to double-resistant bacteria) (Fig 1). The model also contains 80α lytic phage ($P_L$), which can give rise to transducing phage carrying either the erythromycin ($P_E$) or tetracycline ($P_T$) resistance gene. The transducing phage only carry an antibiotic resistance gene, and hence cannot lyse bacteria, but can give rise to $B_{ET}$ bacteria via generalised transduction (Fig 1).

## Antibiotic pharmacodynamics

We added two compartments to the model to track the concentration of antibiotics: $C_E$ for erythromycin, and $C_T$ for tetracycline. The concentrations are expressed in mg/L. We can simulate the addition of a chosen concentration of antibiotic at any given time, and the antibiotics decay at a constant rate $\gamma_E$ and $\gamma_T$ for erythromycin and tetracycline, respectively (Fig 1). The change in concentration of erythromycin over time is therefore expressed as

$$\frac{dC_E}{dt} = -\gamma_E \cdot C_E \tag{1}$$

and the change in concentration of tetracycline over time is expressed as

$$\frac{dC_T}{dt} = -\gamma_T \cdot C_T \tag{2}$$

The antibiotic concentrations are then used to calculate $\varepsilon_{i,j}$, the antibiotic-induced death rate of each antibiotic $i$ ($i \in \{E, T\}$) on each bacterial strain $j$ ($j \in \{E, T, ET\}$), according to a

pharmacodynamic relationship parameterised through Hill Eqs [39,42], expressed as

$$\varepsilon_{i,j} = \mu_j^{max} \cdot \varepsilon_{i,j}^{max} \cdot \frac{C_i^{(H_{i,j})}}{EC50_{i,j}^{(H_{i,j})} + C_i^{(H_{i,j})}} \tag{3}$$

This commonly used function calculates the antibiotic-induced death rate using four parameters: the maximum death rate ($\varepsilon_{i,j}^{max}$), current concentration ($C_i$), half maximal effective concentration ($EC50_{i,j}$), and a Hill coefficient ($H_{i,j}$). Note that $\varepsilon_{i,j}^{max}$ is relative to the maximum growth rate of the corresponding strain $j$ ($\mu_j^{max}$). This was necessary to ensure that these values would work with our previous estimates for bacterial growth, as these were slightly lower than the baseline growth rates estimated here (see below–"Parameter estimation").

## Bacterial growth and phage predation

The bacterial growth rate $\mu_\theta$ ($\theta \in \{E, T, ET\}$, $N = B_E + B_T + B_{ET}$) is modelled using a logistic function given by

$$\mu_\theta = \mu_\theta^{max} \cdot \left(1 - \frac{N}{N^{max}}\right) \tag{4}$$

with a maximum growth rate of $\mu_\theta^{max}$ and carrying capacity $N^{max}$.

Phage predation is modelled as a saturated process. Previous work has suggested that this interaction is more biologically realistic than the more commonly used linear interaction, as it accounts for multiple phage binding to the same bacterium at high phage concentration, leading to a sublinear increase in predation [28,43]. At any given time, the number of phage $F(P_\theta)$ ($\theta \in \{L, E, T\}$), successfully infecting bacteria is calculated according to the Hill function

$$F(P_\theta) = P_\theta \cdot \frac{\beta}{1 + \frac{P_\theta}{P50}} \tag{5}$$

with $\beta$ representing the maximum rate of successful phage adsorption leading to bacterial lysis, and $P50$ corresponding to the phage concentration at half saturation, where the adsorption rate is equal to half the maximum.

In the model, phage burst size $\delta_\theta$ for each bacteria strain $\theta$ ($\theta \in \{E, T, ET\}$) decreases from the maximum phage burst size $\delta^{max}$ as bacterial growth decreases, according to

$$\delta_\theta = \delta^{max} \cdot max\left(0, 1 - \frac{N}{N^{max}} - \frac{\varepsilon_{E,\theta}}{\mu_\theta^{max}} - \frac{\varepsilon_{T,\theta}}{\mu_\theta^{max}}\right) \tag{6}$$

This link between phage burst size and bacterial growth was identified as the most biologically plausible explanation for the dynamics we have previously observed between the bacteria and phage in our system [28]. This decrease happens as bacteria enter stationary phase, when the population approaches carrying capacity $N^{max}$ (as demonstrated previously [28,44,45]), but here the additional effect of antibiotics must be included. To capture this, we use the same effective scaling as the bacterial growth rate (Eq 4) with inclusion of the relative antibiotic-induced death rates $\varepsilon_{E,\theta}$ and $\varepsilon_{T,\theta}$ in the phage burst size estimation. Since antibiotics can kill bacteria, leading to a net negative growth rate, we limit the multiplier value to 0 as a minimum, to prevent a negative burst size.

## Bacteria model equations

The changes in bacteria numbers over time are given by

$$\frac{dB_E}{dt} = \mu_E \cdot B_E - F(P_L) \cdot B_E - F(P_T) \cdot B_E - (\varepsilon_{E,E} + \varepsilon_{T,E}) \cdot B_E \tag{7}$$

{Change in $B_E$ = growth of $B_E$ − infections by $P_L$ − infections by $P_T$ − ery killing − tet killing}

$$\frac{dB_T}{dt} = \mu_T \cdot B_T - F(P_L) \cdot B_T - F(P_E) \cdot B_T - (\varepsilon_{E,T} + \varepsilon_{T,T}) \cdot B_T \tag{8}$$

{Change in $B_T$ = growth of $B_T$ − infections by $P_L$ − infections by $P_E$ − ery killing − tet killing}

$$\frac{dB_{ET}}{dt} = \mu_{ET} \cdot B_{ET} - F(P_L) \cdot B_{ET} + F(P_E) \cdot B_T + F(P_T) \cdot B_E - (\varepsilon_{E,ET} + \varepsilon_{T,ET}) \cdot B_{ET} \tag{9}$$

{Change in $B_{ET}$ = growth of $B_{ET}$ − infections by $P_L$ + infections of $B_T$ by $P_E$ + infections of $B_E$ by $P_T$ − ery killing − tet killing}

Some combinations of phage infection are not explicitly represented in the equations as they represent a flow from one compartment towards the same compartment. $P_L$ are the lytic phage, capable of lysing all the bacteria, and are therefore included in all equations. However, the generalised transducing phage $P_E$ and $P_T$ only carry a bacterial antibiotic resistance gene and cannot lyse bacteria. Whilst they can infect any bacteria in our system to transfer the gene, this process will not affect bacteria already carrying the corresponding gene. For example, if a bacterium $B_E$ is infected by a phage $P_E$, this bacterium remains only resistant to erythromycin, and therefore still belongs to the $B_E$ compartment.

## Phage model equations

The change in phage numbers over time are calculated as

$$\begin{aligned}
\frac{dP_L}{dt} = & [F(P_L) \cdot B_E](t - \tau) \cdot \delta_E \cdot (1 - \alpha) + \\
& [F(P_L) \cdot B_T](t - \tau) \cdot \delta_T \cdot (1 - \alpha) + \\
& [F(P_L) \cdot B_{ET}](t - \tau) \cdot \delta_{ET} \cdot (1 - 2 \cdot \alpha) - \\
& F(P_L) \cdot N - \gamma_P \cdot P_L
\end{aligned} \tag{10}$$

{Change in $P_L$ = new $P_L$ phage from $B_E$ + new $P_L$ phage from $B_T$ + new $P_L$ phage from $B_{ET}$ − $P_L$ phage infecting bacteria − $P_L$ decay}

$$\begin{aligned}
\frac{dP_E}{dt} = & [F(P_L) \cdot B_E](t - \tau) \cdot \delta_E \cdot \alpha + \\
& [F(P_L) \cdot B_{ET}](t - \tau) \cdot \delta_{ET} \cdot \alpha - \\
& F(P_E) \cdot N - \gamma_P \cdot P_E
\end{aligned} \tag{11}$$

{Change in $P_E$ = new $P_E$ phage from $B_E$ + new $P_E$ phage from $B_{ET}$ −

$P_E$ phage infecting bacteria–$P_E$ decay}

$$\frac{dP_T}{dt} = [F(P_L) \cdot B_T](t - \tau) \cdot \delta_T \cdot \alpha + $$
$$[F(P_L) \cdot B_{ET}](t - \tau) \cdot \delta_{ET} \cdot \alpha - $$
$$F(P_T) \cdot N - \gamma_P \cdot P_T$$

(12)

{Change in $P_T$ = new $P_T$ phage from $B_T$ + new $P_T$ phage from $B_{ET}$ −

$P_T$ phage infecting bacteria – $P_T$ decay}

with N = $B_E$+$B_T$+$B_{ET}$. In Eqs 10–12, the latent period $\tau$ corresponds to the delay between bacterial infection and burst, and the transduction probability $\alpha$ corresponds to the proportion of new phage released upon burst which are transducing phage carrying the erythromycin or tetracycline resistance gene. Note that a constant rate $\gamma_P$ is used for phage decay (Eqs 10–12), similar to the rates used for antibiotic decay (Eqs 1–2).

## Parameter estimation

Parameters for bacterial growth and phage predation were originally obtained by fitting our model to *in vitro* data for the same bacteria and phage strains as in this study [28]. Briefly, the parameters were estimated using the Markov chain Monte Carlo Metropolis–Hastings algorithm. We ran the algorithm with two chains until convergence was achieved (determined using the Gelman-Rubin diagnostic, with the multivariate potential scale reduction factor < 1.2 [46]), then for each parameter we generated 50,000 samples from the posterior distributions [28].

Parameters for antibiotic effect were obtained in two steps using the least squares methods for model fitting, which aims to minimise the squared difference between data and model output. Firstly, the antibiotic-induced death rate caused by a specific concentration of antibiotic ($\acute{\varepsilon}$) was obtained by fitting a deterministic growth model given by

$$\frac{dB}{dt} = \mu \cdot \left(1 - \frac{B}{N^{max}}\right) \cdot B - \acute{\varepsilon} \cdot B$$

(13)

to the bacterial concentration B over time, estimated as the mean of three *in vitro* replicates.

$\acute{\varepsilon}$ was then scaled to growth ($\varepsilon = \acute{\varepsilon}/\mu$), to represent the antibiotic-induced death rate relative to bacterial growth rather than an absolute value. This was necessary to ensure that these antibiotic-induced death rates would work with our previous estimates for bacterial growth, as these were slightly lower than the baseline growth rates estimated here. Since we obtained growth curve time series data for 8 concentrations for each strain, we generated 8 estimates of antibiotic-induced death rate for each strain.

Secondly, the three parameters of the Hill equation [39,42] (maximum death rate $\varepsilon^{max}$, Hill coefficient $H$, and half maximum effective concentration *EC50*), which calculates the antibiotic-induced death rate as a function of concentration, were estimated by fitting Eq 3 to the 8 antibiotic effects for each strain.

## Model scenarios considered

**Antibiotics alone.** We start with an environment containing both single-resistant strains at stationary phase ($10^9$ cfu/mL). This mirrors the within-host diversity we could expect to see during bacterial infections [24], and the fact that bacteria most often live at stationary phase in the environment [47]. We first investigate the effect of the presence of either or both erythromycin and tetracycline at concentrations of 1 mg/L, similar to antibiotic concentrations measured *in vivo* during treatment [48,49].

### Antibiotic and phage

We then consider scenarios where phage are present. We first artificially inactivate transduction, by setting the corresponding model parameter to 0. We add $10^9$ pfu/mL of phage, similar to a concentration that could be added during phage therapy, and equivalent to a multiplicity of infection (ratio of phage to bacteria) of 1, similar to what could be found naturally in the environment [1]. We consider scenarios where the phage are present alone, alongside only one antibiotic, or alongside both antibiotics. Phage and antibiotics are assumed to be introduced concurrently at the start, and the simulations run for 24h.

We then repeat these scenarios, but with transduction enabled to levels that were previously observed *in vitro*, with approximately 1 transducing phage carrying an AMR gene generated for each $10^8$ new lytic phage [28].

We repeat the analyses above with either single-resistant strain present at $10^9$ cfu/mL, and the other at $10^6$ cfu/mL (0.1%). This could correspond to a scenario where bacterial diversity is underestimated, with only one type of resistance detected [24].

### Antibiotic and phage level and timing variation

To further investigate the scenario where transduction is enabled and both single-resistant bacterial strains have a starting concentration of $10^9$ cfu/mL, we vary the timing of introduction for antibiotic and phage, with up to 24h delay between their respective additions, as well as varying the concentration of antibiotics between 0.2 and 2.2 mg/L, and phage between $10^5$ and $10^{10}$ pfu/mL, chosen to reflect realistic ranges [1,48,49]. We run these simulations for 48h after the introduction of antibiotics or phage (whichever is added first).

### Phage and bacteria parameters variation

Finally, we vary the probability for phage to perform generalised transduction to identify threshold values which dictate whether multidrug-resistant bacteria appear above detectable levels within 24h. To explore the sensitivity of our results to the parameter values estimated from this single environment, we also conduct a partial rank correlation to assess the effect of varying phage predation parameters (adsorption rate, phage concentration at half saturation, latent period, burst size), bacteria growth rates, and antibiotic and phage decay on total bacteria remaining after 48h, and maximum double-resistant bacteria generated. This allows us to generalise our conclusions as these parameters likely vary for different combinations of phage, antibiotics and bacteria. The ranges evaluated are shown in Table 1, and are either derived from our initial parameterisation of the model [28], or from other studies [50,51].

## Results

### *In vitro* pharmacodynamics of erythromycin and tetracycline

Erythromycin at all concentrations caused a decrease in erythromycin-sensitive NE201KT7 numbers over 6h, but only slowed down growth for erythromycin-resistant NE327 and double-resistant DRPET1, even at 32 mg/L (Fig 2). Tetracycline caused a decrease in bacterial numbers over 6h at concentrations greater than 0.5 mg/L for tetracycline-sensitive NE327, 8 mg/L for tetracycline-resistant NE201KT7, and 4 mg/L for DRPET1 (Fig 2). As a comparison, the minimum inhibitory concentrations (MICs) of erythromycin measured by microbroth dilution were 0.25, >256 and >256 mg/L for NE201KT7, NE327 and DRPET1 respectively. For tetracycline, the MICs were 32, 0.25 and 32 mg/L for NE201KT7, NE327 and DRPET1 respectively.

**Table 1. Model parameter values.** Parameters with no units are dimensionless. All estimates were obtained by fitting the model to *in vitro* data, except those marked with a * which are assumed.

| Name (unit) | | Symbol | Estimate | Range for sensitivity analysis | Reference |
|---|---|---|---|---|---|
| **Bacterial growth parameters** | Carrying capacity (bacteria.mL$^{-1}$) | $N^{max}$ | $2.76 \times 10^9$ | - | [28] |
| | $B_E$ growth rate (h$^{-1}$) | $\mu^{max}_E$ | 1.61 | 1.59–1.63 | |
| | $B_T$ growth rate (h$^{-1}$) | $\mu^{max}_T$ | 1.51 | 1.49–1.53 | |
| | $B_{ET}$ growth rate (h$^{-1}$) | $\mu^{max}_{ET}$ | 1.44 | 1.42–1.47 | |
| **Phage parameters** | Phage adsorption rate (phage$^{-1}$.h$^{-1}$) | $\beta$ | $2.3 \times 10^{-10}$ | $2.1 \times 10^{-10}$–$2.7 \times 10^{-10}$ | |
| | Phage concentration at half saturation (phage.mL$^{-1}$) | P50 | $1.19 \times 10^{10}$ | $1.02 \times 10^{10}$–$1.29 \times 10^{10}$ | |
| | Phage burst size (phage) | $\delta^{max}$ | 50 | 43–54 | |
| | Phage latent period (h) | $\tau$ | 0.61 | 0.60–0.77 | |
| | Transduction probability (phage$^{-1}$) | $\alpha$ | $1.19 \times 10^{-8}$ | $1.11 \times 10^{-8}$–$1.31 \times 10^{-8}$ | |
| | Phage decay rate (h$^{-1}$) | $\gamma_P$ | 0* | 0–0.1 | [50] |
| **Antibiotic parameters** | Erythromycin decay rate (h$^{-1}$) | $\gamma_E$ | 0* | 0–0.1 | [51] |
| | Tetracycline decay rate (h$^{-1}$) | $\gamma_T$ | 0* | 0–0.1 | |
| Erythromycin-induced death rate on $B_E$ | Max death rate (h$^{-1}$) | $\varepsilon^{max}_{E,E}$ | 1.10 | - | This study |
| | Half maximal effective concentration (mg.L$^{-1}$) | EC50$_{E,E}$ | 3.42 | - | |
| | Hill coefficient | $H_{E,E}$ | 0.67 | - | |
| Erythromycin-induced death rate on $B_T$ | Max death rate (h$^{-1}$) | $\varepsilon^{max}_{E,T}$ | 3.91 | - | |
| | Half maximal effective concentration (mg.L$^{-1}$) | EC50$_{E,T}$ | 9.26 | - | |
| | Hill coefficient | $H_{E,T}$ | 0.22 | - | |
| Erythromycin-induced death rate on $B_{ET}$ | Max death rate (h$^{-1}$) | $\varepsilon^{max}_{E,ET}$ | 0.95 | - | |
| | Half maximal effective concentration (mg.L$^{-1}$) | EC50$_{E,ET}$ | 2.47 | - | |
| | Hill coefficient | $H_{E,ET}$ | 0.79 | - | |
| Tetracycline-induced death rate on $B_E$ | Max death rate (h$^{-1}$) | $\varepsilon^{max}_{T,E}$ | 5.40 | - | |
| | Half maximal effective concentration (mg.L$^{-1}$) | EC50$_{T,E}$ | 81.43 | - | |
| | Hill coefficient | $H_{T,E}$ | 0.30 | - | |
| Tetracycline-induced death rate on $B_T$ | Max death rate (h$^{-1}$) | $\varepsilon^{max}_{T,T}$ | 1.66 | - | |
| | Half maximal effective concentration (mg.L$^{-1}$) | EC50$_{T,T}$ | 7.27 | - | |
| | Hill coefficient | $H_{T,T}$ | 2.41 | - | |
| Tetracycline-induced death rate on $B_{ET}$ | Max death rate (h$^{-1}$) | $\varepsilon^{max}_{T,ET}$ | 1.58 | - | |
| | Half maximal effective concentration (mg.L$^{-1}$) | EC50$_{T,ET}$ | 4.42 | - | |
| | Hill coefficient | $H_{T,ET}$ | 1.70 | - | |

The Hill equations fitted well to the antibiotic-induced death rates of varying concentrations of antibiotics on the bacteria (Fig 2, S1 Fig). The corresponding parameter values are presented in Table 1. The antibiotic-induced death rate curves for the DRPET1 are similar to the one for NE201KT7 for tetracycline and NE327 for erythromycin (S1 Fig), which was expected since DRPET1 contains both antibiotic-resistance genes from NE201KT7 (*tetK*) and NE327 (*ermB*).

## Model-predicted antibacterial effect of concurrent erythromycin, tetracycline and bacteriophage presence

When simulating the dynamics of two single-resistant *S. aureus* strains in our mathematical model, as expected, the presence of only one antibiotic at 1 mg/L (4 x MIC for susceptible strains) leads to a decrease in the susceptible strain, while the resistant strain does not decrease (Fig 3A, top row). On the other hand, the presence of both antibiotics, or of only phage without transduction, causes a decrease in both bacterial strains (Fig 3A, top and middle rows).

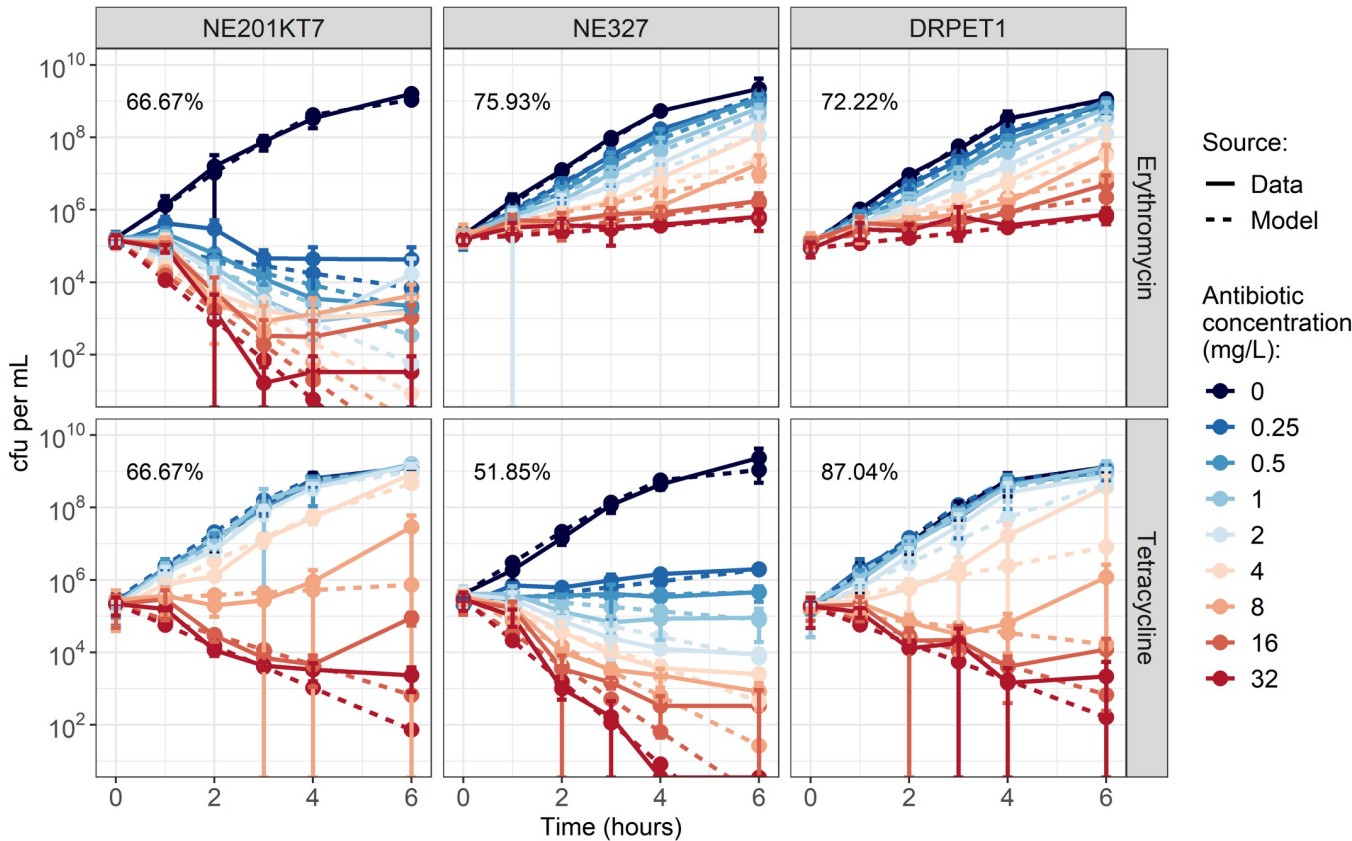

**Fig 2. Growth curves of NE201KT7 (tetracycline-resistant, left), NE327 (erythromycin-resistant, middle) and DRPET1 (double-resistant, right), exposed to varying concentrations of erythromycin (top) or tetracycline (bottom).** Solid lines show *in vitro* data, with error bars indicating mean +/- standard deviation, from 3 replicates. Dashed lines show model output after fitting. Values indicate the percentage of model points that fall within the range of the corresponding *in vitro* data point +/- standard deviation. cfu: colony-forming units. Note that cfu per mL are shown on a log-scale.

The presence of phage and one antibiotic leads to a decrease in both bacterial strains, with the antibiotic-susceptible strain decreasing faster (Fig 3A, middle row). Out of all the conditions shown here, the presence of phage and both antibiotics leads to the fastest decrease in both bacterial strains (Fig 3B). For comparison, to replicate the combined effect of $10^9$ pfu/mL of phage, 1 mg/L of erythromycin, and 1 mg/L of tetracycline, we would need 1.94 mg/L of erythromycin and 1.08 mg/L of tetracycline in the absence of phage (S2 Fig).

Transduction does not appear to affect the antibacterial activity of antibiotics and phage when phage are either present alone, alongside erythromycin, or alongside tetracycline (Fig 3A, bottom row). Double-resistant progeny bacteria ($B_{ET}$) appear, but only reach a maximum concentration of 30 cfu/mL, and do not remain higher than 1 cfu/mL for more than 8h (Fig 3A, bottom row). However, when both erythromycin and tetracycline are present alongside phage capable of transduction, there is a steady increase in the number of $B_{ET}$ throughout 24h, reaching 6 x $10^4$ cfu/mL after 24h (Fig 3B, bottom row and 3B). It is important to note here that when both antibiotics are present, phage numbers do not increase throughout the 24h period, regardless of transduction ability (Fig 3A, middle and bottom rows and 3B, right).

With a reduced starting concentration of either *S. aureus* strain to $10^6$ cfu/mL, while the other remains at $10^9$ cfu/mL, the conclusions are similar as described above, regardless of which strain is in the minority. Fig 4 shows the scenario where tetracycline-resistant bacteria are in minority, and S3 Fig shows the scenario where erythromycin-resistant bacteria are in

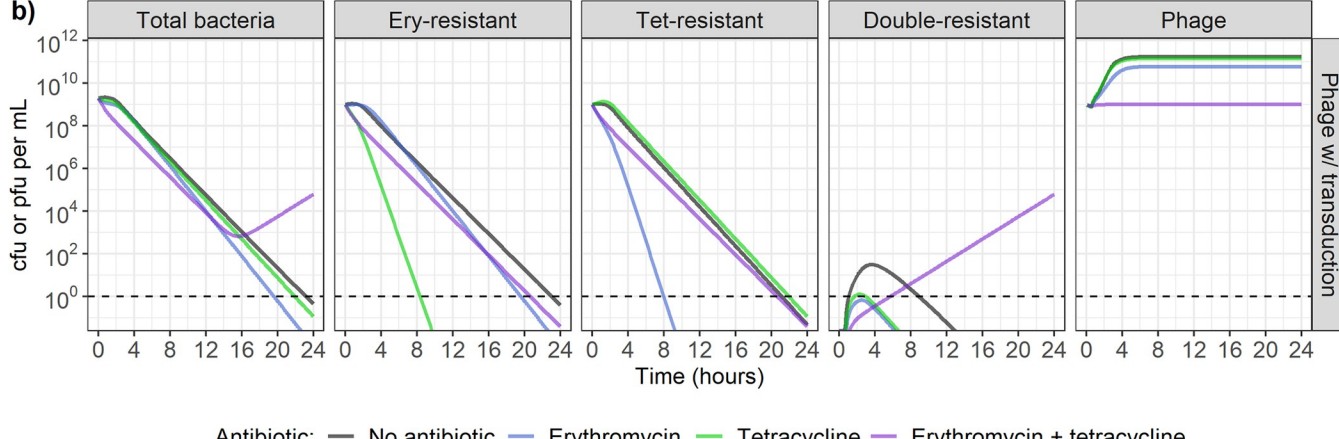

**Fig 3. a) Model-predicted dynamics with two single-resistant strains starting at carrying capacity ($10^9$ colony-forming units (cfu)/mL), in the presence of no antibiotics (1st column), erythromycin only (2nd column), tetracycline only (3rd column), or both erythromycin and tetracycline (4th column), combined with either no phage (top row), phage incapable of transduction (middle row), or phage capable of generalised transduction (bottom row).** The starting strains are either erythromycin-resistant ($B_E$) or tetracycline-resistant ($B_T$). Antibiotics and/or phage ($P_L$) are present at the start of the simulation, at concentrations of 1 mg/L (4 x MIC for susceptible strains) and $10^9$ plaque-forming units (pfu)/mL respectively. Double-resistant bacteria ($B_{ET}$) can be initially generated by generalised transduction only, and then by replication of existing $B_{ET}$. Dashed line indicates the detection threshold of 1 cfu or pfu/mL. **b) Change in bacteria (single-resistant to erythromycin, single-resistant to tetracycline, or double-resistant) and phage numbers depending on the antibiotic exposure, in the presence of phage capable of generalised transduction.** Dashed line indicates the detection threshold of 1 cfu or pfu/mL.

minority. When erythromycin, tetracycline, and phage capable of generalised transduction are all present, $B_{ET}$ are still generated, although they only reach a concentration higher than 1 cfu/mL after 16h, instead of 6h above (Figs 4 and 3).

## Effect of variation in antibiotic and phage timing and concentration on bacterial populations

Under conditions where both erythromycin and tetracycline are present, the timing and concentration of antibiotics and phage capable of generalised transduction drives substantial variation in dynamics, with failure to clear all bacteria after 48h being a possibility (Fig 5A and 5B top row).

With a phage concentration of $10^8$ cfu/mL and for any antibiotics concentration between 0.2 and 2.2 mg/L (Fig 5A), the optimal conditions to clear bacteria within 48h are when phage are initially present, and antibiotics are introduced at least 5h later (Fig 5A, top row). However, this systematically leads to $B_{ET}$ appearance, with a peak concentration between 10 and 100 cfu/mL (Fig 5A, middle row), and a presence time (hours when cfu/mL > 1) of up to 10h (Fig 5A, bottom row). The presence of antibiotics at the same time or shortly before phage leads to failure to clear bacteria within 48h (up to between $10^8$ and $10^{10}$ cfu/mL remaining after 48h - top row), and substantial $B_{ET}$ appearance (maximum concentration up to between $10^8$ and $10^{10}$ cfu/mL—middle row; presence time up to between 30 and 40h- bottom row). Regardless of timings, the addition of at least 2.2 mg/L of antibiotics guarantees that less than 100 bacteria remain after 48h (top row), and no detectable $B_{ET}$ if the antibiotics are added at least 2h before the phage (bottom row).

When keeping the antibiotic concentrations at 1 mg/L, but varying the phage concentration, the impact of the delay between phage and antibiotic presence on the bacterial population is strongly dependent on phage concentration (Fig 5B). We again see that bacteria are not cleared within 48h if antibiotics are present at the same time as or shortly before a concentration of phage between $10^5$ and $10^9$ pfu/mL (Fig 5B, top row). However, we now note that this also occurs if antibiotics are introduced after a concentration of phage between $10^5$ and $10^8$ pfu/mL (Fig 5B, top row). Regardless of timing, the presence of a phage concentration of more than $10^9$ pfu/mL guarantees bacterial clearance within 48h (top row), but leads to $B_{ET}$ if the antibiotics are introduced after the phage (maximum concentration between 10 and 100 cfu/mL—middle row; presence time between 1 and 10h - bottom row).

To investigate the dynamics behind these results, we selected four conditions from Fig 5B (indicated by the 4 black rectangles) with varying antibiotic addition times and plotted the underlying phage and bacteria dynamics over 48h for each (Fig 5C). In all four conditions the starting concentrations are $10^8$ pfu/mL for phage and 1 mg/L for antibiotics, but antibiotics are introduced either 0h, 3h, 5h or 15h after phage. These plots show that if phage are increasing, they stop immediately following antibiotic addition (Fig 5C, 3 and 5h delay). If antibiotics are added too soon after phage (Fig 5C, 0 and 3h delay), phage do not reach a high enough number to exert a sufficient killing pressure on bacteria. In that case $B_{ET}$, which are not substantially affected by either antibiotic, replicate faster than they are killed by phage. After more than 35h, the $B_{ET}$ population reaches a sufficiently high number such that the phage population increases, and the resulting pressure is enough to lead to a net negative bacterial growth rate. If antibiotics are added 15h or later after phage (Fig 5B and 5C), $B_{ET}$ generation will not change, as during this period they will have already arisen by transduction and been removed by phage predation. The presence of antibiotics 5h after phage is optimal to ensure the lowest maximum number of $B_{ET}$ (Fig 5C—compare horizontal dotted lines). This timing allows phage to initially increase to a concentration of $10^{10}$, sufficiently high to exert a strong killing

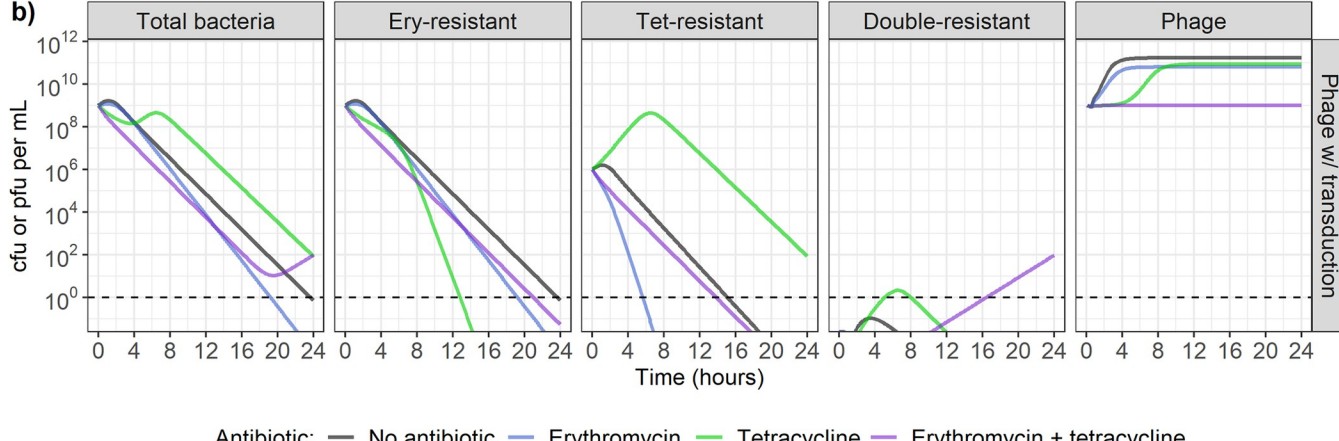

**Fig 4. a) Model-predicted dynamics with one single-resistant strains starting at carrying capacity ($10^9$ colony-forming units (cfu)/mL) and the second in minority ($10^6$ cfu/mL), in the presence of no antibiotics (1st column), erythromycin only (2nd column), tetracycline only (3rd column), or both erythromycin and tetracycline (4th column), combined with either no phage (top row), phage incapable of transduction (middle row), or phage capable of generalised transduction (bottom row).** Erythromycin-resistant bacteria ($B_E$) are initially present at a concentration of $10^9$ cfu/mL, and tetracycline-resistant bacteria ($B_T$) at $10^6$ cfu/mL. Antibiotics and/or phage ($P_L$) are present at the start of the simulation, at concentrations of 1 mg/L (4 x MIC for susceptible strains) and $10^9$ plaque-forming units (pfu)/mL respectively. Double-resistant bacteria ($B_{ET}$) can initially be generated by generalised transduction only, and then by replication of existing $B_{ET}$. Dashed line indicates the detection threshold of 1 cfu or pfu/mL. **b) Change in bacteria (single-resistant to erythromycin, single-resistant to tetracycline, or double-resistant) and phage numbers depending on the antibiotic exposure, in the presence of phage capable of generalised transduction.** Dashed line indicates the detection threshold of 1 cfu or pfu/mL.

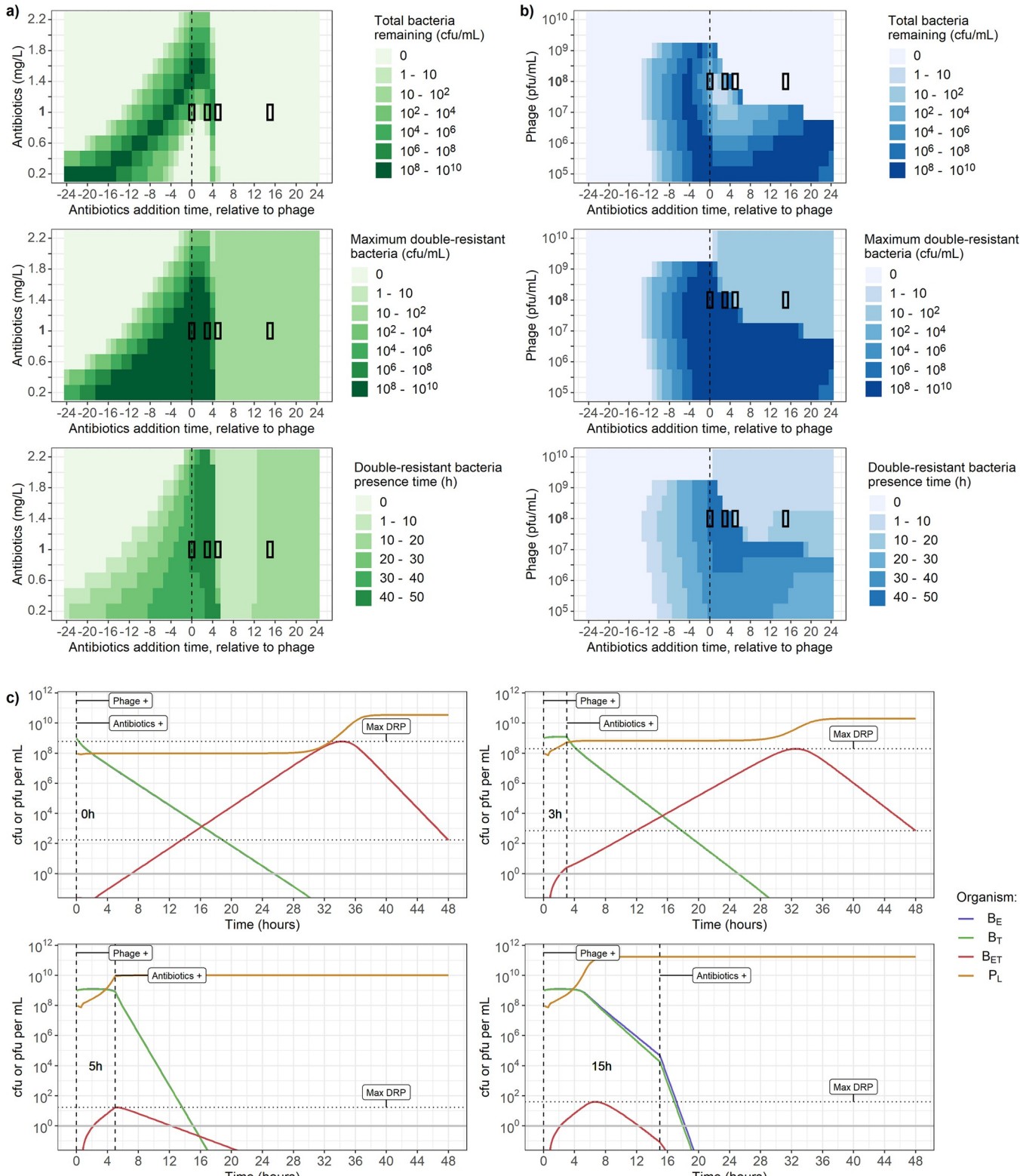

**Fig 5. a-b) Varying timing (x-axis) and dose of antibiotic and phage (y-axis) affects total bacterial count after 48h (top), maximum concentration of double-resistant bacteria ($B_{ET}$) (middle), and time when the concentration of $B_{ET}$ is greater than 1 colony-forming unit (cfu) per mL (bottom). a)** Adding $10^8$ plaque-forming units (pfu) per mL of phage, and between 0.2 and 2.2 mg/L of both erythromycin and tetracycline. **b)** Adding 1 mg/L of both erythromycin and tetracycline, and between $10^5$ and $10^{10}$ pfu/mL of phage. The x-axis indicates the time when antibiotics were added, relative to when phage were added. For

example, the value "4" indicates that phage were present at the start of the simulation, and antibiotics were introduced 4h later. The segments with black borders correspond to the dynamics shown in c). **c) Phage and bacteria dynamics over 48h for 4 conditions taken from panel b.** In all 4 conditions, indicated by the black rectangles, phage are initially present at a concentration of $10^8$ pfu/mL, while erythromycin and tetracycline are both introduced at concentrations of 1 mg/L after either 0h, 3h, 5h or 15h, stated on the plots, with the timing indicated by the vertical dashed lines. Horizontal dotted lines indicate bacteria remaining after 48h (corresponding to the top row of a-b) and maximum double-resistant bacteria ($B_{ET}$) concentration (middle row of a-b). Solid line indicates the detection threshold of 1 cfu or pfu/mL. The concentrations of single-resistant bacteria ($B_E$, blue, and $B_T$, green) overlap and cannot be distinguished.

pressure on bacteria, while the added effect of antibiotics prevents further $B_{ET}$ generation by decreasing the single-resistant strains.

These results also apply to a scenario where one of the two bacterial strains is in the minority (starting concentration of $10^6$ instead of $10^9$ cfu/mL), regardless of which strain is in the minority (S4 & S5 Figs). However, $B_{ET}$ peak, presence time, and total bacteria remaining after 48h decrease faster with a higher dose of antibiotic or phage, or with a higher delay between phage and antibiotics, suggesting that phage and antibiotics are able to exert a greater killing pressure and generate fewer $B_{ET}$ when one strain is in the minority.

## Effect of variation in phage and bacteria parameters on multidrug-resistance evolution

Our results above rely on parameters estimated from phage and bacteria interactions *in vitro*, but these may vary depending on the bacteria, phage, and environment. We can explore these different conditions using our model to quantify the dynamics under varying values for parameters governing phage-bacteria interactions (see Table 1 for the ranges used) to determine whether our results would hold.

We first examine the impact of varying the transduction probability on our results (corresponding to the probability that a transducing phage carrying an AMR gene is released instead of a lytic phage during bacterial burst) as this is a vital and yet poorly quantified parameter. When $10^9$ cfu/mL of each single-resistant strain are simultaneously exposed to $10^9$ pfu/mL of phage and 1 mg/L of erythromycin and tetracycline, varying the transduction probability between $10^{-10}$ and $10^{-6}$ leads to a similar log-fold increase in double-resistant bacteria numbers ($B_{ET}$, Fig 6A). Decreasing the probability only delays the appearance of $B_{ET}$ in the model, and does not prevent it. However, a probability lower than $10^{-11}$ may prevent the appearance of $B_{ET}$ in reality, since the single-resistant strains become almost undetectable ($< 1$ cfu/mL) before the $B_{ET}$ become detectable. If antibiotics are added more than 10h after phage, $B_{ET}$ will have already started declining due to phage predation, hence the antibiotics only contribute to further increasing the decline in bacterial numbers (Figs 5 and 6B). Under these conditions, a transduction probability lower than $10^{-9}$ is necessary to prevent $B_{ET}$ from increasing past the detection threshold (1 cfu/mL) (Fig 6B). If antibiotics are introduced 10h before phage, the resulting decline in single-resistant bacteria prevents any $B_{ET}$ from reaching a detectable level before single-resistant bacteria are eradicated ($< 1$ cfu/mL), even with the highest transduction probability of $10^{-6}$ (Fig 6C).

Looking at how changes in other phage and bacteria parameters may affect our results, partial rank correlation shows that an increase in phage predation either through an increase in phage adsorption rate ($\beta$), phage concentration at half saturation (P50) or phage burst size ($\delta^{max}$) correlates with a decrease in maximum $B_{ET}$ detected over 48h (Fig 6D, blue). For example, the correlation coefficient of -0.95 between $\beta$ and maximum $B_{ET}$ implies that a 100% increase in adsorption rate is correlated with a 95% decrease in maximum double-resistant bacteria detected over 48h. An increase in these parameters is also correlated with a decrease in bacteria remaining after 48h (Fig 6D, red). An increase in latent period ($\tau$), equivalent to a decrease in predation since phage will take longer before lysing the bacteria, is weakly

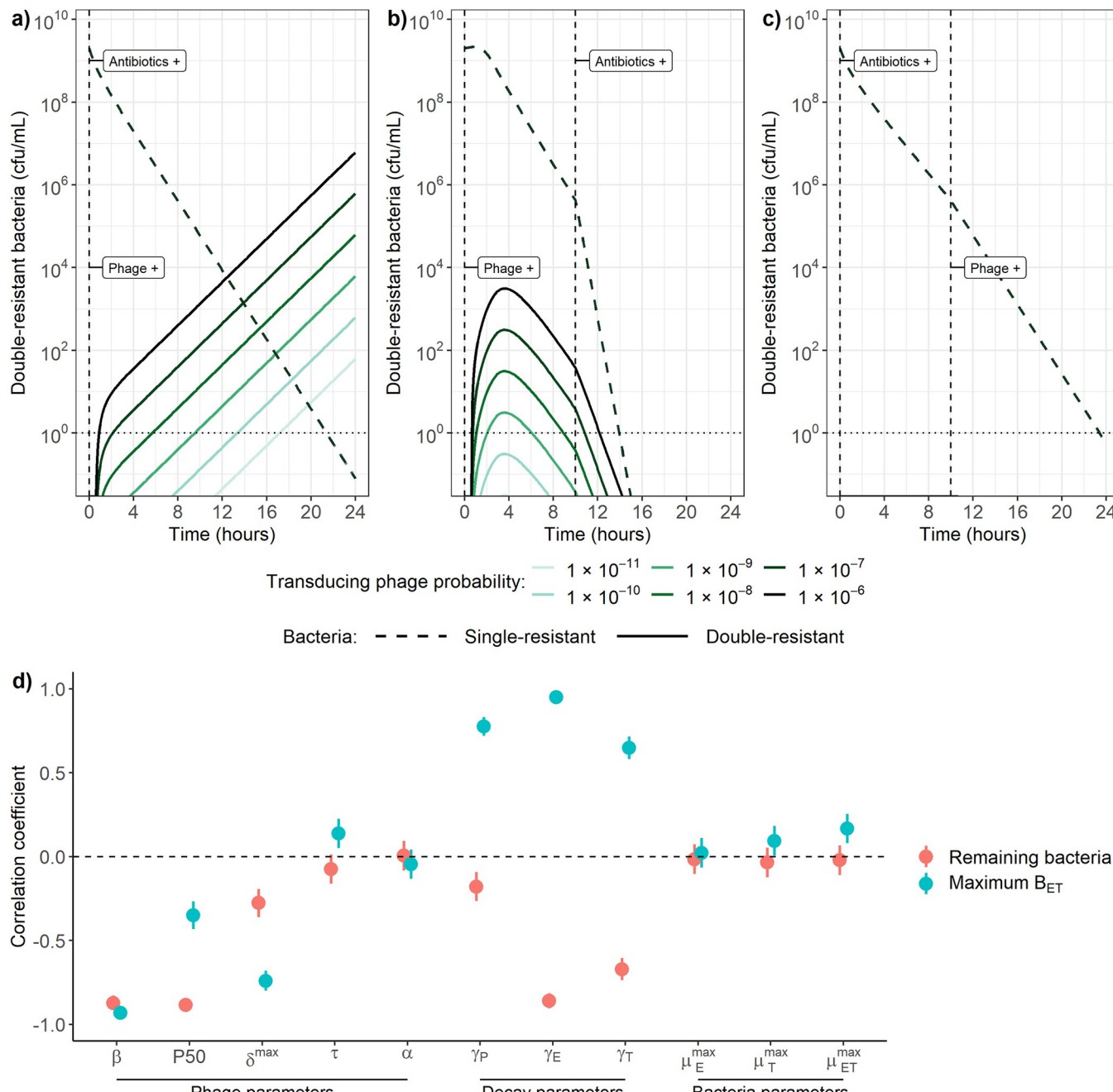

**Fig 6. Sensitivity of phage-bacteria dynamics to changes in model parameters. Effect of varying the transduction probability between $10^{-10}$ and $10^{-6}$ when a) antibiotics and phage are present at the start of the simulation, b) phage are present at the start, antibiotics are introduced 10h later, and c) antibiotics are present at the start, phage are added 10h later.** Transduction probability is defined as the probability that a transducing phage carrying an AMR gene is released instead of a lytic phage during bacterial burst. The dashed lines for single-resistant bacteria overlap and cannot be distinguished. Vertical dashed lines indicate timing of addition of antibiotics or phage. cfu/mL: colony-forming units per mL. **d) Partial rank correlation between model parameters, and remaining bacteria after 48h (pink) or maximum double-resistant bacteria ($B_{ET}$) concentration (blue).** Information on the parameter ranges investigated can be found in Table 1. β: adsorption rate, P50: phage concentration at half saturation, $\delta^{max}$: burst size, τ: latent period, α: transduction probability, $\gamma_P$: phage decay, $\gamma_E$: erythromycin decay, $\gamma_T$: tetracycline decay, $\mu^{max}_E$: $B_E$ growth rate, $\mu^{max}_T$: $B_T$ growth rate, $\mu^{max}_{ET}$: $B_{ET}$ growth rate.

correlated with an increase in maximum $B_{ET}$, but is not substantially correlated with bacteria remaining after 48h. Finally, in this partial rank correlation analysis the transduction probability $\alpha$ was not significantly correlated with either bacteria numbers remaining after 48h or maximum $B_{ET}$, likely since the range investigated was too small (Table 1).

Although we were not able to parameterise our model for antibiotic decay, our 24h time-kill curves suggest that antibiotics no longer decrease bacterial numbers after 24h (S6 Fig). Phage may also be affected by decay in the environment. Hence, we included these parameters in our partial rank correlation analysis. An increase in phage or antibiotic decay ($\gamma_P$, $\gamma_E$, $\gamma_T$) is correlated with an increase in maximum $B_{ET}$ (Fig 6D, blue), but unexpectedly with a decrease in bacteria remaining after 48h (Fig 6D, red). This is explained by the fact that such an increase leads to a weakened killing pressure on $B_{ET}$, which are able to increase faster. This translates to a shorter time before the phage population is able to increase again due to enough bacteria being available for predation, and thus a shorter time before $B_{ET}$ start decreasing due to phage killing (S7 Fig). Finally, single-resistant bacterial growth rates ($\mu^{max}_E$, $\mu^{max}_T$) are not substantially correlated with either maximum $B_{ET}$ or remaining bacteria, while double-resistant growth rate ($\mu^{max}_{ET}$) is only weakly positively correlated with maximum $B_{ET}$, and negatively with remaining bacteria.

## Discussion

### Summary of results

In this work, we reconcile the existing literature suggesting either that phage and antibiotics can synergistically kill bacteria, or that antibiotics reduce the efficacy of phage predation, whilst also considering the synergy of phage and antibiotics on antimicrobial resistance (AMR) evolution. We showed that although phage replication is limited in the presence of antibiotics, which negatively affect bacterial growth, phage are still able to exert a strong killing pressure on bacteria to complement the action of antibiotics. Under such conditions, the concentration and timing of antibiotics and phage are essential: phage introduced after antibiotics at a low concentration may not be able to replicate and hence not contribute to killing the bacterial population. Phage and antibiotics can act synergistically to drive AMR evolution when phage generate multidrug-resistant bacteria by transduction, and antibiotics act as a selection pressure. This is again exacerbated by the reduction in phage replication caused by antibiotics: a low concentration of phage capable of transduction introduced after antibiotics will not exert a strong killing pressure on bacteria, and instead generate multidrug-resistant bacteria at a background rate which are then selected for by the antibiotics.

### Optimal conditions to clear bacteria and minimise AMR evolution

The best conditions to guarantee bacterial eradication within 48h whilst minimising the risk of multidrug-resistant bacteria evolution in our system are when antibiotics are present before phage, and either when erythromycin and tetracycline are present at a concentration of at least 2 mg/L (8 x MIC for susceptible strains), or phage at a concentration of at least $10^{10}$ pfu/mL (Fig 5A and 5B). Multidrug-resistant bacteria generation is also restricted if phage are introduced at least 10h after antibiotics (Fig 5A and 5B). This may be particularly relevant in the context of antibacterial treatment, further discussed below. The worst outcome in which antibiotics and phage fail to rapidly clear all bacteria and instead act synergistically to drive AMR evolution is a combination of a low antibiotic dose ($< 1$ mg/L) with a low phage dose ($< 10^9$ pfu/ml), introduced around the same time (Fig 5A and 5B). Unfortunately, this may correspond to values seen in natural environments where phage are commonly found and antibiotics are residually present due to pollution [2,3].

### Importance of transduction in the environment and during phage therapy

Our results highlight the necessity to better understand the role of transduction in AMR spread and evolution, and not assume by default that it is too rare to be relevant compared to other mechanisms of horizontal gene transfer such as conjugation and transformation. We found that multidrug-resistance evolution remained possible even at the lowest probability we considered here of a transducing phage carrying an AMR gene being released instead of a lytic phage during burst (1 transducing phage per $10^{11}$ lytic phage). Additionally, in this work we assumed that transduction rates are constant, but previous research has shown that sub-MIC antibiotic exposure can lead to an increase in transducing phage, but not lytic ones [52]. Hence, our results may still underestimate the relative impact of transduction versus predation by phage when antibiotics are present. However, our findings are encouraging for conditions under which AMR evolution is limited, since even with a high risk of transduction (1 transducing phage per $10^6$ lytic phage), if phage are only introduced to an environment more than 10h after antibiotics, multidrug-resistant bacteria do not reach detectable levels before single-resistant strains are eradicated (Fig 6C).

To the best of our knowledge, our model is the first to consider the potential consequences of transduction in a system where concentrations of bacteria, phage, and antibiotic are similar to those we may see during phage therapy. Previously our work only considered bacteria and phage capable of generalised transduction in the absence of antibiotics [28], whilst others have explored bacteria phage and antibiotics without including generalised transduction [6–8,10]. We echo the conclusions from these previous studies which highlighted that the timing of antibiotics and phage introduction during phage therapy can affect the rate at which bacteria are cleared, but extend these to show that timing may also impact the risk for multidrug-resistant bacteria to be generated. Here, we suggest that, although both sequential treatments can ultimately lead to bacterial eradication, the timing leads to a trade-off between a slower clearing rate of bacteria (if antibiotics are added before phage), and a higher risk of multidrug-resistance evolution (if phage are added before antibiotics). Future studies and clinical trials of phage therapy should investigate varying timings of phage and antibiotics, instead of only investigating their simultaneous application, and consider the risk of transduction during treatment.

In any case, our ability to measure the impact of transduction as a driver of AMR evolution *in vivo* is currently limited since individuals are not routinely screened for phage. A first step to measure this despite the limitation may be to investigate evidence for within-patient changes in the resistance profile of *S. aureus* isolates, as these would likely be caused by transduction [23]. In the case of antibiotic treatment, the natural presence of phage capable of transduction may explain instances of treatment failure, if these generate multidrug-resistant strains which are then selected for by the antibiotics. Future studies monitoring therapeutic outcomes of antibacterial treatment in patients where phage are also detected will be essential to better understand how our findings translate to *in vivo* settings.

### Limitations

The major limitation of our work is the deterministic nature of our model. While it does not account for stochastic events which would play a large role when bacterial numbers are low, the deterministic model is useful for analysis purposes, as it represents the average scenario we would observe. In reality, we would likely see either bacterial clearance or unexpected increases at low numbers of bacteria. We only generated model predictions for up to 48h, as our parameter values were obtained using data from experiments over 24h. Beyond this time, bacteria and phage may be affected negatively due to resource depletion, depending on the environment.

Our model does not include some dynamics which may be present *in vivo*, as we do not currently have robust data available to parameterise these features, and would instead have had to rely on assumptions or previously estimated parameter values from different settings. Notably, we have not included the effect of the immune system, which may limit the number of multidrug-resistant bacteria generated as it could suppress both bacteria and phage populations *in vivo* [53,54]. If the model was extended to include the immune system, it would also have to consider potential detrimental effects of large doses of phage and antibiotics, which would restrict these concentrations to prevent side effects *in vivo* [55,56].

In addition, we assume that all the bacteria in our environment are equally susceptible to phage infection, and have not considered the possibility for any further evolution. This includes adaptation mutation allowing multidrug-resistant bacteria to overcome any fitness cost, as well as resistance to phage. In our own previous *in vitro* experiments using these *S. aureus* strains and phage, we did not detect evidence of bacterial resistance to phage appearing within 24h [28]. However, such resistance may arise over longer periods of time [57,58]. The threat represent by phage resistance evolution is clear for multidrug-resistant bacteria, as these would then no longer be affected by either antibiotics nor phage. On the other hand, if single antibiotic-resistant strains developed resistance to phage in our system, they would still be removed by the other antibiotic they are susceptible to, but may then be more likely to benefit from transduction and become multidrug-resistant, depending on the phage resistance mechanism. If resistance occurs by targeting and neutralising phage DNA in the cell (e.g. CRISPR--Cas [58]), bacterial DNA injected by a transducing phage would not be targeted, hence protecting the resistant bacteria from phage lytic infection whilst still allowing transduction. However, if resistance occurs by preventing of phage binding (e.g. surface receptor modification [58], although there is limited evidence of this type of phage resistance in *S. aureus* [59]), then the bacteria would no longer benefit from transduction. In any case, the risk of phage resistance could be mitigated by rapid killing of bacteria, before phage resistance arises, highlighting the importance of measuring how fast bacterial eradication occurs.

Although we varied the concentration of antibiotics in our results, we have consistently added erythromycin and tetracycline in equal amounts. Our model would allow us to change this, yet we have chosen not to for simplicity and because the antibacterial effect curves look similar for these two antibiotics in our setting (S1 Fig). However, for other antibiotics it may be necessary to revisit this assumption and investigate concentrations which may better reflect those to which bacteria are exposed to in the environment or during antibacterial treatment.

## Generalisability

Our model is extensively parameterised using data from a single phage and three *S. aureus* strains, making it a robust tool to study the dynamics of these organisms, as it relies on a minimum number of assumptions [28]. However, the parameters we have estimated (adsorption rate, phage concentration at half saturation, burst size, latent period and transduction probability) will likely vary depending on the phage, bacteria, and environment studied. Our sensitivity analysis shows that the model outputs are reasonable with alternative parameter values, predicting for example that phage with a higher predation capacity (higher adsorption rate, phage concentration at half saturation or burst size, or lower latent period) would clear more bacteria within 48h, and reduce the maximum number of multidrug-resistant bacteria generated. This model has been developed as part of an interdisciplinary project alongside *in vitro* experiments, hence it could be easily re-parameterised using data for other strains of bacteria and phage showing similar dynamics of lysis and generalised transduction. The structure of the model is generalisable to other systems of generalised transducing phage and bacteria, as it

captures the relevant biological characteristics of phage predation and generalised transduction [28].

## The unique dynamics of phage, bacteria, and antibiotics

We suggest that transduction and the effect of antibiotics should be considered in the context of the previously described unique dynamics of phage and bacteria. These imply that phage must first reach a certain concentration (previously referred to as "inundation threshold") before they can offset bacterial growth and decrease the bacterial population, and bacteria must first reach a certain concentration ("proliferation threshold") before the phage population can increase [10]. Generalised transduction and antibiotics affect the size of the bacterial population, and therefore how phage interact with bacteria to clear them and generate multidrug-resistant bacteria. Thus, multidrug-resistant bacteria are able to increase in our model if phage are initially present at a concentration lower than the inundation threshold (Fig 5C). This also explains our counterintuitive observation that higher decay rates may lead to less bacteria remaining after 48h (Fig 6D), as this would allow bacteria to reach the proliferation threshold sooner, and therefore allow phage to increase up to the inundation threshold sooner (S7 Fig). Importantly, our results similarly suggest that higher decay rates for antibiotics present alongside phage would reduce bacteria remaining after 48h, at the cost of a higher peak concentration of multidrug-resistant bacteria, since this decay would allow bacteria to reach the proliferation threshold sooner (Fig 6C, S7 Fig). This knowledge may be further useful in the context of phage therapy, to select the antibiotics that will be given alongside phage [51].

## Conclusions

Our results demonstrate the complex synergy between phage and antibiotics to kill bacteria and drive the evolution of AMR. We suggest this synergy leads to a trade-off between a slower clearing rate of bacteria (if antibiotics are added before phage), and a higher risk of multidrug-resistance evolution (if phage are added before antibiotics), further exacerbated by low concentrations of either phage or antibiotics. Interdisciplinary frameworks such as ours combining *in vitro* data and mathematical models are key to understanding both fundamental AMR evolution, and new interventions like phage therapy or screening for phage in patients. Our conclusions form hypotheses to guide future experimental and clinical work, notably for studies of phage therapy which should consider the risk for multidrug-resistance evolution by transduction, and investigate varying timings and concentrations of phage and antibiotics instead of only their simultaneous use.

## Supporting information

**S1 Fig. Antibiotic-induced death rate erythromycin and tetracycline measured *in vitro* (pink) and obtained after fitting Hill equations (blue).** Death rate is relative to bacterial growth, such that a value greater than 1 indicates killing (net negative growth), while a value between 0 and 1 indicates only a decrease in growth rate. NE201KT7 contains a tetracycline-resistance gene (*tetK*), NE327 contains an erythromycin-resistance gene (*ermB*) and DRPET1 contains both resistance genes. The Hill equation is shown in Eq 3.
(TIF)

**S2 Fig. a) The antibacterial effect of 1 mg/L of both erythromycin and tetracycline alongside $10^9$ pfu/mL of phage is equivalent to b) the effect of 1.97 mg/L of erythromycin and 1.08 mg/L of tetracycline in the absence of phage.** This was estimated by setting the

concentration of phage to 0 in b) and fitting the concentrations of erythromycin and tetracycline to reproduce the decrease in bacteria numbers seen in a). cfu: colony-forming units; pfu: plaque-forming units.
(TIF)

**S3 Fig. a) Model-predicted bacterial dynamics in the presence of no antibiotics (1st column), erythromycin only (2nd column), tetracycline only (3rd column), or both erythromycin and tetracycline (4th column), combined with either no phage (top row), phage incapable of transduction (middle row), or phage capable of generalised transduction (bottom row).** Tetracycline-resistant bacteria ($B_T$) are initially present at a concentration of $10^9$ colony-forming units (cfu)/mL, and erythromycin-resistant bacteria ($B_E$) at $10^6$ cfu/mL. Antibiotics and/or phage ($P_L$) are present at the start of the simulation, at concentrations of 1 mg/L and $10^9$ plaque-forming units (pfu)/mL respectively. Double-resistant bacteria ($B_{ET}$) can be generated by generalised transduction only. Dashed line indicates the detection threshold of 1 cfu or pfu/mL. **b) Change in bacteria (single-resistant to erythromycin, single-resistant to tetracycline, or double-resistant) and phage numbers depending on the antibiotic exposure, in the presence of phage capable of generalised transduction.**
(TIF)

**S4 Fig. Effect of varying antibiotic and phage timing and concentration when the tetracycline-resistant bacterial strain ($B_T$) is in minority ($10^6$ cfu/mL). a-b) Varying timing (x-axis) and dose of antibiotic and phage (y-axis) affects total bacterial count after 48h (top), maximum concentration of double-resistant bacteria ($B_{ET}$) (middle), and time when the concentration of $B_{ET}$ is greater than 1 colony-forming unit (cfu) per mL (bottom). a)** Adding $10^8$ plaque-forming units (pfu) per mL of phage, and between 0.2 and 2.2 mg/L of both erythromycin and tetracycline. **b)** Adding 1 mg/L of both erythromycin and tetracycline, and between $10^5$ and $10^{10}$ pfu/mL of phage. The x-axis indicates the time when antibiotics were added, relative to when phage were added. For example, the value "4" indicates that phage were present at the start of the simulation, and antibiotics were introduced 4h later. The segments with black borders correspond to the dynamics shown in c). **c) Phage and bacteria dynamics over 48h for 4 conditions taken from panel b.** In all 4 conditions, indicated by the black rectangles, phage are initially present at a concentration of $10^8$ pfu/mL, while erythromycin and tetracycline are both introduced at concentrations of 1 mg/L after either 0h, 3h, 5h or 15h, stated on the plots, with the timing indicated by the vertical dashed lines. Horizontal dotted lines indicate bacteria remaining after 48h (corresponding to the top row of a-b) and maximum double-resistant bacteria ($B_{ET}$) concentration (middle row of a-b). Solid line indicates the detection threshold of 1 cfu or pfu/mL.
(TIF)

**S5 Fig. Effect of varying antibiotic and phage timing and concentration when the erythromycin-resistant bacterial strain ($B_E$) is in minority ($10^6$ cfu/mL). a-b) Varying timing (x-axis) and dose of antibiotic and phage (y-axis) affects total bacterial count after 48h (top), maximum concentration of double-resistant bacteria ($B_{ET}$) (middle), and time when the concentration of $B_{ET}$ is greater than 1 colony-forming unit (cfu) per mL (bottom). a)** Adding $10^8$ plaque-forming units (pfu) per mL of phage, and between 0.2 and 2.2 mg/L of both erythromycin and tetracycline. **b)** Adding 1 mg/L of both erythromycin and tetracycline, and between $10^5$ and $10^{10}$ pfu/mL of phage. The x-axis indicates the time when antibiotics were added, relative to when phage were added. For example, the value "4" indicates that phage were present at the start of the simulation, and antibiotics were introduced 4h later. The segments with black borders correspond to the dynamics shown in c). **c) Phage and bacteria**

**dynamics over 48h for 4 conditions taken from panel b.** In all 4 conditions, indicated by the black rectangles, phage are initially present at a concentration of $10^8$ pfu/mL, while erythromycin and tetracycline are both introduced at concentrations of 1 mg/L after either 0h, 3h, 5h or 15h, stated on the plots, with the timing indicated by the vertical dashed lines. Horizontal dotted lines indicate bacteria remaining after 48h (corresponding to the top row of a-b) and maximum double-resistant bacteria ($B_{ET}$) concentration (middle row of a-b). Solid line indicates the detection threshold of 1 cfu or pfu/mL.
(TIF)

**S6 Fig. Growth curves of NE201KT7 (tetracycline-resistant, left), NE327 (erythromycin-resistant, middle) and DRPET1 (double-resistant, right), exposed to varying concentrations of erythromycin (top) or tetracycline (bottom).** The minimum inhibitory concentration values for bacteria at 24h were identical to the ones for stock bacteria, suggesting that antibiotic decay rather than acquired resistance is responsible for the increase in bacteria numbers after 24h. Error error bars indicate mean +/- standard deviation, from 3 replicates. cfu: colony-forming units. Note that cfu per mL are shown on a log-scale.
(TIF)

**S7 Fig. Impact of phage and antibiotic decay on phage and bacteria dynamics over 48h.** The conditions shown are: no decay (a), phage decay (b), erythromycin decay (c), and tetracycline decay (d). In all 4 conditions, phage and antibiotics (erythromycin and tetracycline) are initially present at concentrations of $10^9$ pfu/mL and 1 mg/L respectively. Rates of decay are set to either 0 or 0.1 per hour.
(TIF)

# Acknowledgments

The authors would like to thank Caroline Memmi for helpful discussions on antibiotic bio-availability and *in vivo* antibiotic concentrations, as well as Joseph Standing and his research group for advice regarding antibiotic pharmacodynamics modelling.

# Author Contributions

**Conceptualization:** Quentin J. Leclerc, Jodi A. Lindsay, Gwenan M. Knight.

**Data curation:** Quentin J. Leclerc.

**Formal analysis:** Quentin J. Leclerc.

**Investigation:** Quentin J. Leclerc.

**Methodology:** Quentin J. Leclerc, Jodi A. Lindsay, Gwenan M. Knight.

**Resources:** Jodi A. Lindsay.

**Software:** Quentin J. Leclerc.

**Supervision:** Jodi A. Lindsay, Gwenan M. Knight.

**Validation:** Jodi A. Lindsay, Gwenan M. Knight.

**Visualization:** Quentin J. Leclerc, Jodi A. Lindsay, Gwenan M. Knight.

**Writing – original draft:** Quentin J. Leclerc.

**Writing – review & editing:** Quentin J. Leclerc, Jodi A. Lindsay, Gwenan M. Knight.

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
