## [Decision Letter · Decision Letter 0]

25 May 2022

Dear Mr Leclerc,

Thank you very much for submitting your manuscript "Modelling the synergistic effect of bacteriophage and antibiotics on bacteria: killers and drivers of resistance evolution" for consideration at PLOS Computational Biology.

As with all papers reviewed by the journal, your manuscript was reviewed by members of the editorial board and by several independent reviewers. In light of the reviews (below this email), we would like to invite the resubmission of a significantly-revised version that takes into account the reviewers' comments.

We cannot make any decision about publication until we have seen the revised manuscript and your response to the reviewers' comments. Your revised manuscript is also likely to be sent to reviewers for further evaluation.

Sincerely,

Eric Lofgren, MSPH, PhD

Associate Editor

PLOS Computational Biology

Thomas Leitner

Deputy Editor

PLOS Computational Biology

Reviewer's Responses to Questions

**Comments to the Authors:**

Reviewer #1: This an interesting study, investigating how timing of antibiotic and phage combination against Staph aureus affects bacterial clearance and multi-AMR evolution through horizontal gene transfer (generalised transduction) using deterministic numerical simulations. The results are intuitive: the application of antibiotics prior to phage reduces the killing power of phages but reduces the chance of multi-AMR, and vice versa if phages are applied first. The models are parameterised from in vitro data collected here and in previous work, but a wider range of parameter values are also explored. The findings are therefore likely to be relevant and reasonably generalisable.

1. While interesting, the novelty of the work is perhaps marginal. The study is very similar to their previous (ref 23), but with timing of phage/antibiotic treatment altered. I would like the authors to be more upfront about this.

2. Please mention lateral transduction as another mechanism of transduction. I saw a Penades paper cited, but could not find direct reference to this important phenomenon.

3. Slightly odd to use a lysogenic phage as a phage therapy model, but the authors show that it is mainly lytic in these lab conditions. That said, bacteria typically evolve very rapid resistance lytic phage. Please discuss and speculate on impact on population and AMR evolution dynamics.

Reviewer #2: review is uploaded as an attachment

Reviewer #3: `Modelling the synergistic effects of bacteriophage and antibiotics on bacteria...' by Leclerc, Lindsay and Knight is a combined experimental and theoretical study into the impact of bacteriophage-mediated gene transfer (transduction) on the development of antimicrobial resistance in bacteria. The authors perform population dynamics experiment on S. aureus; they use these experiments to fit the parameters of a mathematical model, which they then use to explore a much wider parameter space. The main finding of this research is that, in their model, bacteriophage-mediated transduction can significantly increase the probability of multidrug-resistant bacterial mutants arising.

The conclusions of this article will be of interest to the wide community interested in combination phage/antibiotic therapy because they highlight the importance of generalised transduction in generating multidrug resistant bacteria. The general presentation of the work is very clear. However, there are serious deficiencies in the presentation and clarity of the model that make it difficult for me to address the validity of these conclusions and I recommend this paper only be considered again after major revisions.

There are some serious errors and ambiguities in the model presentation. It is sometimes unclear to me what the model is and I am doubtful as to whether it represents what the authors want it to. Some of the criticisms below probably don't make sense because I have misunderstood what the authors are trying to do, but that is the point: I will need to see a revised manuscript to properly assess the model, as I cannot understand it yet.

-Equations should be parts of sentences and have terminal punctuation. They should not be referred to before they are presented.

-Variables like \\mu{max} and \\epsilon{max} look unwieldy and would be clearer with the max in superscript or some other notation.

-Eq. 3 should have the fraction written out fully, not y/x.

-Eq. 4 appears to have the (\\psi_L+\\psi_T)*B_E repeated; similarly for Eq. 5. Once this term is repeated with a prefactor of \\mu_E (units of inverse hours) and once without, so this does not work dimensionally.

-Asterisks are unnecessary to show multiplication and different parentheses {[()]} should be used to make nested expressions clearer, e.g., in Eq. 4.

-In the eqs on p11 it's not obvious why certain terms are included or not, e.g., I would expect all phages to kill all bacteria, but only selected combinations seem to be represented. Likewise, it is not clear which terms correspond to lysis of bacteria by phage and what corresponds to transduction.

-Transduction requires that the transduced bacterium is not lysed by the phage and this lysis failure presumably also happens with bacteria that are not transduced. There should therefore be an overall lysis failure rate included.

-It is very helpful to have each term specified in words below these equations, as you do. We also need a clearer explanation of these equations, addressing the ambiguities raised above.

-\\phi_{\\theta} is defined over a page after it is first introduced and I don't think the definition is correct: \\phi_{\\theta} is presumably the probability that the bacterium is infected by the phage specified by \\theta. If you want the probability of infection by at least one phage, that would be something like 1-\\product_{\\theta}(1-\\phi_\\theta).

Assuming that the model is correct, the conclusions are interesting, and I appreciate how the authors have pinned down the many parameters in their model by the use of original experiments rather than relying on literature data. The sensitivity analysis is helpful, but we need to know rather more about how you fitted the parameters to the experiment. What fitting methods did you use and how did you (or did you) ensure that the fitted parameters were robust, i.e., can you adjust pairs or groups of parameters to give a similarly good fit to the model? This model has a large number of parameters and so I would want more information to be convinced that there is no qualitatively different set of parameters that will give as good a fit to the data. I understand that some of this work has already been done in ref. [23] but it needs to be summarized here too. I would also like the authors to explicitly state what the current work adds to reference [23] as this is not clear to me. Fig. 1 seems almost identical to a figure in ref. [23] and it would be helpful to acknowledge that here.

The authors use various phrases to describe generalised transduction that seem to attribute some intentionality to the bacteriophage and bacteria, e.g., `accidentally' (l42, l65), `by mistake' (l67) and `a mispackaging and thus a biological error similar to a mutation' (l88). I object to this for two reasons: i) The whole of evolution is an `accident' in the same sense, so it's illogical to single out generalised transduction in this way and ii) These phrases would seem to imply that the transduction process gives no selective advantage for the bacteria or bacteriophage. As transduction generates drug-resistant bacteria, and hence more resilient hosts for the bacteriophage, this implicit claim at least needs some detailed, explicit justification. I suggest removing these and other similar phrases.

The data in Fig. 5 a-b is rather hard to read and there is no `red line' (l400) on these figures or at least it is not visible in my pdf. A clearer way to present this could be as a heat map, enabling the authors to systematically explore two control parameters simultaneously. The numbers on the axes of this figure, particularly the exponents, are too small and I recommend splitting this up into two figures.

**Have the authors made all data and (if applicable) computational code underlying the findings in their manuscript fully available?**

Reviewer #1: Yes

Reviewer #2: Yes

Reviewer #3: Yes

PLOS authors have the option to publish the peer review history of their article (what does this mean?). If published, this will include your full peer review and any attached files.

Reviewer #1: No

Reviewer #2: No

Reviewer #3: No
---

## [Decision Letter · Decision Letter 1]

12 Sep 2022

Dear Mr Leclerc,

Thank you very much for submitting your manuscript "Modelling the synergistic effect of bacteriophage and antibiotics on bacteria: killers and drivers of resistance evolution" for consideration at PLOS Computational Biology.

As with all papers reviewed by the journal, your manuscript was reviewed by members of the editorial board and by several independent reviewers. In light of the reviews (below this email), we would like to invite the resubmission of a significantly-revised version that takes into account the reviewers' comments.

We cannot make any decision about publication until we have seen the revised manuscript and your response to the reviewers' comments. Your revised manuscript is also likely to be sent to reviewers for further evaluation.

Sincerely,

Eric Lofgren, MSPH, PhD

Academic Editor

PLOS Computational Biology

Thomas Leitner

Section Editor

PLOS Computational Biology

Reviewer's Responses to Questions

**Comments to the Authors:**

Reviewer #1: You have done a very good job responding to reviewer comments. Thank you for producing an interesting paper.

Reviewer #3: The authors have made considerable efforts to improve the article, e.g., Fig. 5 is now much clearer, and they have addressed most of my concerns. However, although I now have a clearer picture of the model, I don't believe it is a coherent model, either in the implementation here, or in the authors' previous paper, ref. 28, doi:10.1128/msystems.00135-22. I am not optimistic that the authors will be able to resolve these issues without further major revisions so I would recommend that the paper be rejected.

Specifically, the authors mix discrete and continuous equations in an ad hoc fashion without sufficient justification. If Eq. 8 is supposed to be a differential equation rather than a difference equation it shouldn't explicitly reference the `time-step', which is 1/omega. If it's a difference equation, it should be displayed as such. Also, I'm puzzled as to what the authors mean by a `time-step' here at all. It is clear from the figures that the numerical integration is performed more frequently than once per hour, whereas omega=1/hour.

I agree with the authors about the problem: if the `timestep' is very long then the number of bacteria can become negative. There are two ways to deal with this: one is to use a much smaller time step and then, if necessary, deal with very small bacterial numbers by imposing an arbitrary cut off (e.g., if the total bacterial number in the system drops below one, that number could be set to zero). The other way of dealing with this, which the authors attempt, is to add correction terms to give the correct probabilities of infection and growth within much longer time steps. However, the authors do this in an ad hoc fashion, by just adding the term containing omega to Eq. 8. This step is not justified and I am not convinced that they have kept all terms to lowest order in omega, or that higher order terms (omega^2 etc.) are not needed with such a long time step.

In any future version, I would expect the authors to start from an appropriate differential equation model and derive their difference equation model (if that is what it is) formally, justifying all steps mathematically and making sure that the meaning of all terms (such as the `timestep') is clearly defined, and it is clear how they are related, e.g., how is the delay between infection and lysis related to the `timestep'?

Also, when describing Eq. 3 it would be clearer to describe epsilon_{i,j} as the antibiotic-induced death rate (or similar) rather than just the `effect' of the antibiotic. I struggled for a bit because I assumed that `effect' should be a factor that is multiplicative on the growth rate.

**Have the authors made all data and (if applicable) computational code underlying the findings in their manuscript fully available?**

Reviewer #1: Yes

Reviewer #3: Yes

PLOS authors have the option to publish the peer review history of their article (what does this mean?). If published, this will include your full peer review and any attached files.

Reviewer #1: No

Reviewer #3: No
---

## [Editor Report · Decision Letter 2]

17 Nov 2022

Dear Dr Leclerc,

We are pleased to inform you that your manuscript 'Modelling the synergistic effect of bacteriophage and antibiotics on bacteria: killers and drivers of resistance evolution' has been provisionally accepted for publication in PLOS Computational Biology.

Best regards,

Eric Lofgren, MSPH, PhD

Academic Editor

PLOS Computational Biology

Thomas Leitner

Section Editor

PLOS Computational Biology

---

## [Editor Report · Acceptance letter]

25 Nov 2022

PCOMPBIOL-D-22-00368R2 

Modelling the synergistic effect of bacteriophage and antibiotics on bacteria: killers and drivers of resistance evolution

Dear Dr Leclerc,

I am pleased to inform you that your manuscript has been formally accepted for publication in PLOS Computational Biology. Your manuscript is now with our production department and you will be notified of the publication date in due course.

With kind regards,

Marianna Bach
